# The Diagnostic Accuracy of Second Trimester Plasma Glycated CD59 (pGCD59) to Identify Women with Gestational Diabetes Mellitus Based on the 75 g OGTT Using the WHO Criteria: A Prospective Study of Non-Diabetic Pregnant Women in Ireland

**DOI:** 10.3390/jcm11133895

**Published:** 2022-07-04

**Authors:** Delia Bogdanet, Michelle Toth Castillo, Helen Doheny, Louise Dervan, Miguel-Angel Luque-Fernandez, Jose A. Halperin, Paula M. O’Shea, Fidelma P. Dunne

**Affiliations:** 1College of Medicine, Nursing and Health Sciences, School of Medicine, National University of Ireland, H91 TK33 Galway, Ireland; louise.dervan@health.wa.gov.au (L.D.); fidelma.dunne@nuigalway.ie (F.P.D.); 2Divisions of Haematology, Brigham and Women’s Hospital, Harvard Medical School, Boston, MA 02115, USA; mtothcastillo@partners.org (M.T.C.); jhalperin@bwh.harvard.edu (J.A.H.); 3Department of Clinical Biochemistry, Saolta University Health Care Group (SUHCG), Galway University Hospitals, H91 YR71 Galway, Ireland; helen.doheny@hse.ie (H.D.); paulam.oshea@hse.ie (P.M.O.); 4Department of Epidemiology, Harvard T.H. Chan School of Public Health, Boston, MA 02115, USA; miguel-angel.luque@lshtm.ac.uk; 5Department of Epidemiology and Population Health, London School of Hygiene and Tropical Medicine, London KT1 2EE, UK

**Keywords:** gestational diabetes, pregnancy, biomarker, prediction

## Abstract

The aim of this study was to evaluate the ability of second trimester plasma glycated CD59 (pGCD59), a novel biomarker, to predict the results of the 2 h 75 g oral glucose tolerance test at 24–28 weeks of gestation, employing the 2013 World Health Organisation criteria. This was a prospective study of 378 pregnant women. The ability of pGCD59 to predict gestational diabetes (GDM) was assessed using adjusted ROC curves for maternal age, BMI, maternal ethnicity, parity, previous GDM, and family history of diabetes. The pGCD59 levels were significantly higher in women with GDM compared to women with normal glucose tolerance (*p* = 0.003). The pGCD59 generated an adjusted AUC for identifying GDM cases of 0.65 (95%CI: 0.58–0.71, *p* < 0.001). The pGCD59 predicted GDM status diagnosed by a fasting glucose value of 5.1 mmol/L with an adjusted AUC of 0.74 (95%CI: 0.65–0.81, *p* < 0.001). Analysis of BMI subgroups determined that pGCD59 generated the highest AUC in the 35 kg/m^2^ ≤ BMI < 40 kg/m^2^ (AUC: 0.84 95%CI: 0.69–0.98) and BMI ≥ 40 kg/m^2^ (AUC: 0.96 95%CI: 0.86–0.99) categories. This study found that second trimester pGCD59 is a fair predictor of GDM status diagnosed by elevated fasting glucose independent of BMI and an excellent predictor of GDM in subjects with a very high BMI.

## 1. Introduction

Gestational diabetes (GDM) is defined as “any degree of glucose intolerance with onset or first recognition during pregnancy” [1]. According to the International Diabetes Federation (IDF), in 2019, 16.8 million pregnancies were affected by GDM [2]. GDM is a major cause of maternal and foetal adverse pregnancy outcomes. It is well established, however, that GDM treatment reduces the risk of these perinatal complications [3]. Therefore, it is of utmost importance that GDM is diagnosed accurately.

GDM is currently diagnosed by either a one-step procedure—the 2 h 75 g oral glucose tolerance test (OGTT) [4,5,6] or a two-step procedure—a 1 h 50 g glucose challenge test (GCT) followed by a 3 h 100 g OGTT in women who failed the GCT [7]. There is a wealth of information in the research literature documenting that the measurement of plasma glucose is susceptible to errors, unreliable and affected by sampling conditions [8,9]. In addition to the sampling factors that can influence the glucose test result, the OGTT is a long and unpleasant test that requires fasting, the ingestion of a glucose drink that may result in vomiting and is associated with significant direct and indirect costs for the woman and the healthcare service. International organizations now recognize that there is a need for a more robust, single, non-fasting test to replace the OGTT in the diagnosis of GDM [10].

CD59, a key complement inhibitor, is an 18–20 kDa glycosyl-phosphatidylinositol-anchored cell membrane glycoprotein that prevents the formation of the membrane attack complex (MAC) and thereby complement-mediated cell damage and lysis [11,12,13,14]. CD59 is ubiquitously expressed in mammalian cells; a soluble form detached from cell membranes is present in blood and urine [15,16]. Exposure to hyperglycaemia leads to the non-enzymatic glycation of the protein forming glycated CD59 (pGGC59), which is functionally inactive thus increasing MAC-mediated cell damage and lysis upon complement activation [17,18].

Preliminary work on the potential of pGCD59 as a biomarker for the screening and diagnosis of GDM showed promising results. The pGCD59 assessment during two-step GDM screening, revealed higher pGCD59 levels in women who failed the GCT and the 3 h 100 g OGTT (Carpenter and Coustan criteria [19]) compared to controls [20].

The aim of this study was to investigate the diagnostic accuracy of second trimester pGCD59 in identifying GDM cases in a one-step approach using a 2 h 75 g OGTT at 24–28 weeks of gestation (WG) adjudicated based on the 2013 World Health Organisation (WHO) criteria.

## 2. Materials and Methods

The study protocol was published in [21]. Consecutive pregnant women attending their first antenatal visit at Galway University Hospital, Galway, Ireland, were prospectively recruited between November 2018 and March 2020. All pregnant women over 18 years old were invited to participate in the study except women with pre-established diabetes. The patient information leaflet was given at the first antenatal appointment and a member of the research team explained the purpose of the study and its methodology. If agreeable, a consent form was signed.

The weight and height of the women were measured at the first antenatal visit using SECA scales model 799 (22089 Hamburg Germany) and the BMI was calculated and stratified according to WHO guidelines as underweight (<18.5 kg/m^2^), normal weight (18.5–24.9 kg/m^2^), overweight (25–29.9 kg/m^2^), and obese (≥30 kg/m^2^) [22]. Peripheral maternal blood pressure was measured using an ambulatory blood pressure monitor (SECA mVSA 535). Women had an ultrasound scan to confirm gestational age.

Women underwent routine second-trimester screening for GDM (24–28 WG). Each woman who attended was advised to fast for 8–12 h prior to attending for the test. Participants were not advised regarding any carbohydrate or exercise restriction for the days preceding the test. On the day of the test, each participant was given Rapilose OGTT Solution (Penlan Healthcare Ltd., Abbey House, Wellington Way, Weybridge, UK) which comes in liquid form and is available in a ready-to-use 300 mL pouch containing 75 g anhydrous glucose and instructed not to eat or drink anything further for the duration of the test. Patients were also instructed not to smoke. Blood samples were taken fasting and at 1 h and 2 h after the ingestion of glucose. Gestational diabetes was defined by one abnormal plasma glucose value in the OGTT according to the WHO criteria (fasting value 5.1 mmol/L (92 mg/dL), 1 h value 10 mmol/L (180 mg/dL), and 2 h value 8.5 mmol/L (153 mg/dL)) [4]. Whole blood was drawn in fluoride oxalate specimen tubes for plasma glucose measurement, and glucose was measured on the Roche Cobas 8000 analyser using the hexokinase method (Roche Diagnostics, Indianapolis, IN, USA).

Blood (10 mL) was taken into ethylenediaminetetraacetic acid (EDTA) for pGCD59 measurement at the first antenatal visit, at the time of routine blood testing and at the time of routine 2 h 75 g OGTT. Each pGCD59 plasma sample was separated into two 500 µL barcoded aliquots and kept at −80 °C. All laboratory specimens were given a coded identity number to maintain participant confidentiality. A clinical database linked to the barcoded samples was developed and pseudo-anonymised. This data was encrypted, password protected, and kept on a secure server. After the recruitment process was completed, an aliquot of each participant’s EDTA plasma sample was transported on dry ice to the Laboratory for Translational Research, Haematology Division, Department of Medicine, Brigham and Women’s Hospital, Boston, USA for pGCD59 analysis. The pGCD59 was determined using the enzyme-linked immunosorbent (ELISA) test previously published by Ghosh et al. [23]. The intra-assay coefficient of variation (CV) was 3.0%. Test operators were blind to the women’s glucose tolerance status.

A clinical database linked to the barcoded samples was developed and pseudo-anonymised. The constructed database contained baseline clinical information (age, weight, height, ethnicity, blood pressure, week of gestation), obstetric history (parity, gravida), lifestyle variables (smoking status, alcohol consumption) and laboratory data (OGTT results, pGCD59 levels, date of sampling) on each patient.

### 2.1. Statistical Analysis

The power calculation and sample size have been previously described [21].

For continuous variables, mean and standard deviations/median and interquartile range were used, while for categorical variables, count/percentages were used. We used the χ^2^ test for categorical variables, the Wilcoxon–Mann–Whitney test for continuous variables not normally distributed, and Student *t*-tests for continuous variables normally distributed to compare the baseline characteristics of pregnant women with normal glucose tolerance (NGT) to the baseline characteristics of pregnant women who developed GDM. Delta pGCD59 (ΔpGCD59) was calculated by the difference in pGCD59 levels between the first and second trimester of pregnancy.

Unadjusted nonparametric receiver operating characteristic (ROC) curves and adjusted ROC curves for maternal age, BMI, maternal ethnicity, parity, previous GDM, and family history of diabetes were used to evaluate the ability of pGCD59 to predict the results of the 24–28 WG OGTT. Then, their respective area under the curve (AUC) was calculated with their 95%CI.

Missing data were assumed to be completely at random and a complete case analysis was performed. To explore the missing data mechanisms and verify the plausibility of the missing at random assumption, we checked if the proportion of missingness in the variables differed between the levels of the normal glucose tolerance group and GDM groups. This was carried out using summary statistics and performing a hypothesis test of an association between missingness in variables and the outcomes.

In all analyses, *p* < 0.05 was considered statistically significant. SPSS for Windows, version 20, was used for all statistical analyses (IBM SPSS Statistics for Windows, version 20 SPSS, Chicago, IL, USA).

### 2.2. Ethics

Ethical approval for this study was granted by the Clinical Research Ethics Committee, Galway, Ireland (Reference No- C.A. 2026).

## 3. Results

We enrolled 2037 people in the study, 7 of whom withdrew, 11 had miscarriages, and 2 underwent pregnancy termination (TOP). Anaemia (*n* = 1), cystic fibrosis (*n* = 1), and needle fear (*n* = 5) were among the reasons for withdrawal (Figure 1). Among the remaining 2017 participants, 230 women were diagnosed with GDM. Of all the GDM mothers, 42 did not have samples taken at both the first antenatal appointment and at the time of the OGTT, resulting in a total of 188 GDM participants. A total of 376 study participants with NGT matched for age, BMI, and ethnicity who had samples taken at the first antenatal appointment and at the time of the OGTT were included in the study, resulting in a total cohort of 564 study participants. For the aims of this study, we only included participants with the first sample (Table 1) taken at <14 WG, a second sample (T2) and an OGTT taken at weeks 24–28 WG and singleton pregnancy. Of the 564 subjects, 378 participants’ samples met the sampling time inclusion criteria (NGT *n* = 275, GDM *n* = 103) (Figure 1).

Table 1 lists the characteristics of the participants as well as their laboratory test results. Study participants with GDM had higher SBP (*p* = 0.03) and mean BP (*p* = 0.02) compared to women with NGT. There were no additional differences in the two cohorts’ baseline characteristics. T2 pGCD59 levels were significantly higher in women with GDM (*p* = 0.003). ΔpGCD59 was significantly higher in women with NGT compared to the GDM cohort (*p* = 0.01). Women with GDM had higher glucose levels at all time points on the OGTT, as expected.

T2 pGCD59 generated an unadjusted AUC for predicting GDM at 24–28 WG of 0.58 (data not shown). After adjustment for baseline characteristics, the AUC increased to 0.65 (95%CI: 0.58–0.71) (Figure 2A). We explored T2 CD59 prediction of GDM status diagnosed by individual values on the OGTT. The pGCD59 predicted GDM status diagnosed with a fasting glucose value of 5.1 mmol/L with an adjusted AUC of 0.74 (95%CI: 0.65–0.81) (Figure 2B), a fasting glucose value of 5.3 mmol/L with an adjusted AUC of 0.75 (95%CI: 0.64–0.84) (Figure 2C), a 1 h glucose value of 10 mmol/L with an adjusted AUC of 0.63 (95%CI: 0.54–0.72) (Figure 2D) and a 2 h glucose value of 8.5 mmol/L with an adjusted AUC of 0.66 (95%CI: 0.55–0.76). We further explored the ability of ΔpGCD59 to predict GDM status based on the overall OGTT result and individual OGTT values (Figure 3). The results were similar to the predictive values of pGCD59 alone.

We further investigated the ability of pGCD59 to predict GDM status in BMI subcategories (Figure 4). BMI was stratified: BMI < 25 kg/m^2^ (NGT *n* = 109, GDM *n* = 31); 25 ≤ BMI < 30 kg/m^2^ (NGT *n* = 80, GDM *n* = 34); 30 ≤ BMI < 35 kg/m^2^ (NGT *n* = 56, GDM *n* = 25); 35 ≤ BMI < 40 kg/m^2^ (NGT *n* = 22, GDM *n* = 9); BMI ≥ 40 kg/m^2^ (NGT *n* = 8; GDM *n* = 4). pGCD59 generated the highest AUC in the 35 kg/m^2^ ≤ BMI < 40 kg/m^2^ (AUC: 0.84 95%CI: 0.69–0.98) and BMI ≥ 40 kg/m^2^ (AUC: 0.96 95%CI: 0.86–0.99) categories.

## 4. Discussion

In this prospective study, we analysed the ability of second trimester pGCD59 taken at the same time as the 2 h 75 g OGTT (24–28 WG) to predict GDM status diagnosed using the 2013 WHO criteria. We found that pGCD59 is a fair predictor of GDM status diagnosed by elevated fasting glucose independent of BMI and an excellent predictor of GDM in subjects with a very high BMI.

In recent years, pGCD59 has shown promising results as a biomarker for glucose intolerance. Measurements of pGCD59 correctly identified subjects with type 2 diabetes with a sensitivity of 93%, specificity of 100% and AUC of 0.98 [23,24] and showed a 2-week turnover rate response to diabetes treatment. This rapid turnover rate is of particular importance in GDM.

Ghosh et al. [20] showed that pGCD59 predicted abnormal GCT results with an adjusted AUC of 0.92 (95%CI 0.88–0.93) and predicted GDM by the 3 h 100 g OGTT with an adjusted AUC of 0.92 (95%CI 0.77–0.91), independent of age, BMI, ethnicity of history of diabetes. While the pGCD59 samples were taken at a similar time as in our study (second trimester, 24–28 WG) there are several potential factors that can account for the difference in pGCD59 predictive ability between the two studies. In the Ghosh et al., study, women were initially screened with a GCT—if the 1 h glucose levels were between 7.8 (140 mg/dL)–10.6 mmol/L (190 mg/dL) then the woman proceeded to a 3 h 100 g OGTT. The diagnosis of GDM was based on the Carpenter and Coustan diagnosis criteria (fasting glucose 5.3 mmol/L (95 mg/dL), 1 h 10.1 mmol/L (180 mg/dL), 2 h 8.7 mmol/L (155 mg/dL), 3 h 7.8 mmol/L (140 mg/dL); two or more values met or exceeded are required to make the diagnosis). Using this approach, fewer women are diagnosed as having GDM as higher diagnostic values identify women with more complicated forms of GDM and, arguably, women with milder forms of GDM are not captured nor are those with an isolated abnormal fasting glucose. Previous studies support this conclusion [25,26,27]. This would explain the higher median pGCD59 values in the GDM group in the Ghosh paper compared with ours (3.23 vs. 2.6 SPU). Furthermore, the study by Ghosh et al. was more ethnically diverse compared with our study and it also included multiple pregnancies which increase the risk of GDM [28,29]. All these differences make a comparison between studies difficult. The lower risk population and inclusion of milder GDM cases are a possible explanation for the lower AUC found in our study compared to the Ghosh study. Furthermore, when we restricted to a higher risk population (high BMI), the AUC increased significantly. This is further supported by the study by Ma et al. [30] who assessed pGCD59 ability to predict GDM in a high-risk population (BMI ≥ 29 kg/m^2^) diagnosed by a 75 g 2 h OGTT and WHO criteria. Despite the earlier timing of pGCD59 sampling (<20 weeks), the team found the pGCD59 can predict a diagnosis of GDM (<20 WG) with an AUC of 0.86 (95%CI: 0.83–0.90). While the differences in sampling and diagnosis timings impede the potential of a direct comparison between our study and the study by Ma. et al., both studies found that pGCD59 had a good prediction capacity for GDM in women with an elevated BMI.

In our cohort, pGCD59 predicted a diagnosis of GDM based on elevated fasting glycaemia with a higher AUC compared with the prediction based on the 1 h and 2 h glucose levels. As a glycated protein, pGCD59 might better reflect background glucose levels compared to responsive glycaemia after glucose ingestion. Another explanation is that elevated fasting glucose levels alone identify a majority of women that will be diagnosed with GDM [31,32,33]. The Hyperglycaemia and Adverse Pregnancy Outcome (HAPO) Study found that 55% of women would be diagnosed with GDM by using the fasting glucose threshold for diagnosis alone [34].

Our study also examined the ability of ΔpGCD59 to predict GDM based on the assumption that the change in pGCD59 levels between the first and second trimester of pregnancy would better reflect the progression of glycemia during pregnancy between women with GDM and NGT compared with pGCD59 taken at the time of the OGTT alone. We found no difference in GDM predictive capacity between ΔpGCD59 and second trimester pGCD59. A possible explanation for this is that even in normal pregnancy there is a natural increase in glucose levels even if to a lesser extent than GDM pregnancies [35]. Therefore, it is plausible that the difference in glycaemic levels between the first two trimesters of pregnancy as reflected by ΔpGCD59 is not superior to second trimester pGCD59 alone.

This study has several limitations. The lack of ethnic diversity reflects our current population but limits the generalizability of the results. Another limitation is the lack of information on gestational weight gain. The COVID-19 pandemic has had a major impact on the scientific community, including this research. We had to reassess and alter the study design due to recurrent lockdowns, laboratory closures and reopening, difficulties in procuring laboratory consumables, and limited staff availability. As a result, the study deviated from the original published protocol [21] and the number of samples analysed was reduced. Lastly, due to the small number of cases in each BMI subgroup, the results of our sensitivity analysis should be interpreted with caution.

While pGCD59′s ability to predict GDM at 24–28 WG in a general population was less than expected, this study found that pGCD59 can predict elevated fasting glucose levels at 24–28 WG with reasonable accuracy. Additionally, we found that pGCD59 can identify women with GDM with excellent accuracy when we restricted the analysis to a subcohort with very high BMI. It appears that pGCD59 performs very well in identifying GDM in high-risk populations and in cohorts with more complicated forms of GDM. Further studies are required to confirm pGCD59’s ability to identify GDM in much larger cohorts that would allow for complex subcohort analysis and would further our knowledge into the role of pGCD59 in GDM and factors that influence pGCD59 levels (such as ethnicity, variable glycaemic thresholds, metabolic variables). Future research exploring the link between pGCD59 and GDM diagnosis in cohorts screened for GDM using the 75 g 2 h OGTT employing the WHO criteria would allow for a meaningful and direct comparison between studies.

## Figures and Tables

**Figure 1 jcm-11-03895-f001:**
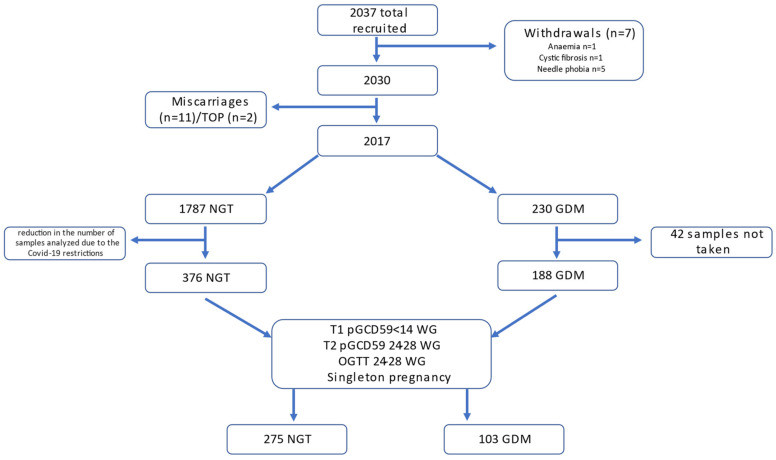
Study flowchart.

**Figure 2 jcm-11-03895-f002:**
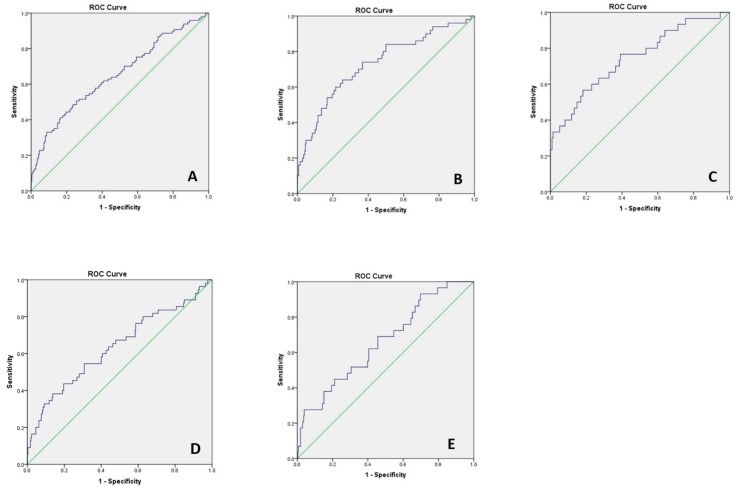
pGCD59 (24–28 WG)—adjusted ROC curves for maternal age, BMI, maternal ethnicity, parity, previous GDM and family history of diabetes at Galway University Hospital, Galway, Ireland between November 2018 and March 2020, *n* = 378. (**A**) pGCD59 prediction of GDM status AUC:0.65 95%CI: 0.58–0.71; (**B**) pGCD59 prediction of fasting glucose of 5.1 mmol/L, AUC: 0.74 95%CI: 0.65–0.81; (**C**) pGCD59 prediction of fasting glucose of 5.3 mmol/L, AUC: 0.75 95%CI: 0.64–0.84; (**D**) pGCD59 prediction of 1 h glucose of 10 mmol/L AUC 0.63 95%CI: 0.54–0.72; (**E**) pGCD59 prediction of 2 h glucose of 8.5 mmol/L, AUC:0.66 95%CI: 0.55–0.76.

**Figure 3 jcm-11-03895-f003:**
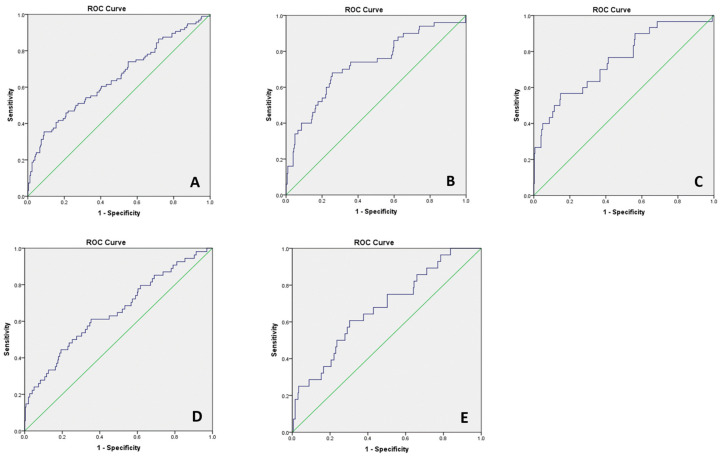
ΔpGCD59—adjusted ROC curves for maternal age, BMI, maternal ethnicity, parity, previous GDM and family history of diabetes at Galway University Hospital, Galway, Ireland between November 2018 and March 2020, *n* = 378. (**A**) ΔpGCD59 prediction of GDM status AUC:0.65 95%CI: 0.58–0.71; (**B**) ΔpGCD59 prediction of fasting glucose of 5.1 mmol/L, AUC:0.73 95%CI: 0.64–0.81; (**C**) ΔpGCD59 prediction of fasting glucose of 5.3 mmol/L, AUC:0.75 95%CI: 0.65–0,84; (**D**) ΔpGCD59 prediction of 1 h glucose of 10 mmol/L AUC 0.65 95%CI: 0.56–0.73; (**E**) ΔpGCD59 prediction of 2 h glucose of 8.5 mmol/L, AUC:0.67 95%CI: 0.56–0.77.

**Figure 4 jcm-11-03895-f004:**
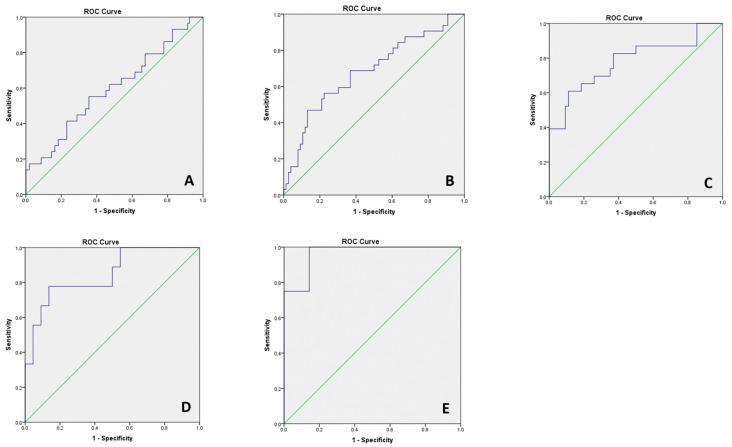
pGCD59 (24–28 WG) prediction of GDM status (24–28 WG) by BMI categories—adjusted ROC curves for maternal age, BMI, maternal ethnicity, parity, previous GDM, and family history of diabetes at Galway University Hospital, Galway, Ireland between November 2018 and March 2020, *n* = 378. (**A**) BMI < 25 m/kg^2^, AUC 0.60 95%CI: 0.47–0.71; (**B**) 25 kg/m^2^ ≤ BMI < 30 kg/m^2^, AUC 0.68 95%CI: 0.56–0.79; (**C**) 30 kg/m^2^ ≤ BMI < 35 kg/m^2^, AUC 0.78 95%CI: 0.65–0.90; (**D**) 35 kg/m^2^ ≤ BMI < 40 kg/m^2^, AUC 0.88 95%CI: 0.69–0.98; (**E**) BMI ≥ 40 kg/m^2^ AUC 0.96 95%CI: 0.86–0.99.

**Table 1 jcm-11-03895-t001:** Women’s baseline characteristics and laboratory values at Galway University Hospital, Galway, Ireland between November 2018 and March 2020, *n* = 378.

Baseline Characteristics	NGT *n* = 275 (IQR/%)	GDM *n* = 103 (IQR/%)	*p* Value
Age (years)	33.6 (31.1–36.4)	34.8 (31.7–37.4)	0.07
WG at booking	12.7 (12–13.1)	12.4 (12–13.1)	0.60
Gravida	2 (1–3)	2 (1–3)	0.30
Parity	1 (0–1)	1 (0–2)	0.60
Height (cm)	165 (161.6–169.5)	164 (160–169)	0.10
Weight (kg)	73 (64–86.2)	75.7 (64.2–89.6)	0.30
BMI (kg/m^2^)	26.4 (23.3–31)	28.7 (23.7–31.9)	0.10
Ethnicity (white)	243/275 (88.4)	88/103 (85.4)	0.60
SBP (mmHg)	120 (112–126)	122 (114–130)	0.03
DBP (mmHg)	69 (62–75)	69 (64–79)	0.10
Mean BP (mmHg)	86 (80.6–91)	88.3 (79.6–94.3)	0.02
WG at delivery	40 (39–40.8)	39.4 (38.8–40.4)	0.07
Alcohol at booking	4/275 (1.4)	1/103 (0.9)	0.20
Alcohol before pregnancy	233/275 (84.7)	81/103 (78.6)	0.30
Non-smoker	145/275 (52.7)	55/103 (53.3)	0.90
Smoker at booking visit	14/275 (5)	8/103 (7.7)	0.20
Laboratory values			
T2pGCD59 (SPU)	2.39 (1.85–2.9)	2.6 (1.9–3.4)	0.003
ΔpGCD59	1.2 (0.4–2)	1.1 (0.09–1.7)	0.01
OGTT 24–28 weeks:			
Fasting glucose (mmol/L)	4.4 (4.2–4.6)	5.1 (4.6–5.3)	<0.01
1 h glucose (mmol/L)	7 (5.8–7.9)	10 (8.7–10.8)	<0.01
2 h glucose (mmol/L)	5.5 (4.8–6.4)	7.1 (6–8.7)	<0.01
Mean glucose (mmol/L)	5.5 (5.1–6.1)	7.2 (6–8.7)	<0.01

BMI: body mass index; BP: blood pressure; DBP: diastolic blood pressure; GDM: gestational diabetes; NGT: normal glucose tolerance; OGTT: oral glucose tolerance test; SBP: systolic blood pressure; Table 1: 1st trimester; WG: weeks of gestation. The data is presented as median values and interquartile ranges (IQR) for continuous variables and the number of cases and percentage of cases out of total (%) for dichotomous variables.

## Data Availability

Data is available upon reasonable request.

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
