# Peer review of "The Diagnostic Accuracy of Second Trimester Plasma Glycated CD59 (pGCD59) to Identify Women with Gestational Diabetes Mellitus Based on the 75 g OGTT Using the WHO Criteria: A Prospective Study of Non-Diabetic Pregnant Women in Ireland"

_jcm, 2022, doi:10.3390/jcm11133895_

Round 1
Reviewer 1 Report
The study is well designed and written, and the findings have certain clinical significance.
I suggest the authors briefly describe the study design and study protocol in this study, instead of just referring to another study, as it is very inconvenient.
Why the study only focused on the second trimester? How about the performance of pGCD59 in the first and/or third trimester?
The findings showed that pGCD59 has certain predictive value for pregnant women with high BMI. I suggest the authors also report the GDM rate in subjects with such BMI.
Reviewer 2 Report
GDM is defined as any impaired glucose tolerance, which appeared for the first time during pregnancy or was then identified. Due to the expected increase in the prevalence of GDM, it seems crucial to diagnose it in proper way. As the authors rightly point out, the current diagnosis of gestational diabetes is based on OGTT. This test is often poorly tolerated by pregnant patients, associated with the need to sit for two hours and stressing the carbohydrate metabolism. Therefore, it seems reasonable to search for new tests for the diagnosis of GDM.
I would like to suggest the following points for the improvement of the manuscript:
1. It would be worth briefly describing how 75g OGTT is performed, noting that 3 blood glucose measurements should be performed.
2. Please complete the citation in the line 50 in the introduction
3. Line 97: “using a the enzyme-…” – typos?
4. Line 116: p < 0.05 (not P)
5. Please standardize the abbreviation throughout the manuscript (pGCD59 not a pgCD59 f.e. in line 112)
6. Table 1 - please describe the abbreviation IQR. Please describe the numbers in table 1, whether they are mean / medians and in parentheses, minimum and maximum values
7. Proofreading of the manuscript for fixing typos is recommended
I recommend accepting the article for publication after minor corrections have been made. In general, the work is very interesting and well written.
Reviewer 3 Report
The study tried to investigate the diagnostic accuracy of second trimester glycated CD59 (pGGC59) in identifying GDM cases in a 1-step approach using a 2-h 75g OGTT.
My comments are the following
Comment 1. Page 3, line 83: “2-h value 8.5 mmol/L (154 mg/dl))”. The 2-h value is 153 instead of 154, please correct it.
Comment 2: Page 3, line 101: “The power calculation and sample size have been previously described”. By assessing your previously published protocol, I noticed that in the sample size calculation, there is a description of the numbers of patients visiting the department, describing how many patients could be included in the study. However, there are no presenting data regarding the calculation of the required sample needed, to answer your hypothesis.
Comment 3: Page 3, line 115: “Missing data were assumed to be completely at random and a complete case analysis was performed”. Did the authors perform Little's test of missing completely at random? Otherwise, how did they assume that the data were missing completely at random?
Comment 4: Page 3, lines 128-129: “Of the 1787 NGT participants, a total of 376 NGT participants who had samples taken at the 1st antenatal appointment and at the time of the OGTT were included in the study”. How did those patients were selected? Potential source of selection bias.
Comment 5: Page 3, lines 130-132: “Of the 564 subjects, we selected only the participants with the first sample (T1) taken at <14 WG, a second sample (T2) and an OGTT taken at weeks 24-28 WG and singleton pregnancy (NGT n=275, GDM n= 103)”. According to the study design and the published protocol of the study, all women should have had a first trimester sample and sample at the timing of OGTT. If there were patients with missing data (only one of the two samples taken) please report the number of those patients, the reason of the missing data and how did you handle them statistically. Is the derived sample sufficient to test the research hypothesis?
Comment 6: Page 7, lines 175-176: “We further investigated the ability of pGCD59 to predict GDM status in BMI subcategories”. Was the sample size adequate to proceed with the subgroup analyses?
Round 2
Reviewer 3 Report
This is an improved version of the manuscript. I have no further comments.